# Accuracy Assessment in Convolutional Neural Network-Based Deep Learning Remote Sensing Studies—Part 1: Literature Review

**Aaron E. Maxwell** *,† , **Timothy A. Warner** † and **Luis Andrés Guillén**

Department of Geology and Geography, West Virginia University, Morgantown, WV 26505, USA; Tim.Warner@mail.wvu.edu (T.A.W.); lg0018@mix.wvu.edu (L.A.G.)
* Correspondence: Aaron.Maxwell@mail.wvu.edu; Tel.: +1-304-293-2026
† These authors contributed equally to this work.

**Abstract:** Convolutional neural network (CNN)-based deep learning (DL) is a powerful, recently developed image classification approach. With origins in the computer vision and image processing communities, the accuracy assessment methods developed for CNN-based DL use a wide range of metrics that may be unfamiliar to the remote sensing (RS) community. To explore the differences between traditional RS and DL RS methods, we surveyed a random selection of 100 papers from the RS DL literature. The results show that RS DL studies have largely abandoned traditional RS accuracy assessment terminology, though some of the accuracy measures typically used in DL papers, most notably precision and recall, have direct equivalents in traditional RS terminology. Some of the DL accuracy terms have multiple names, or are equivalent to another measure. In our sample, DL studies only rarely reported a complete confusion matrix, and when they did so, it was even more rare that the confusion matrix estimated population properties. On the other hand, some DL studies are increasingly paying attention to the role of class prevalence in designing accuracy assessment approaches. DL studies that evaluate the decision boundary threshold over a range of values tend to use the precision-recall (P-R) curve, the associated area under the curve (AUC) measures of average precision (AP) and mean average precision (mAP), rather than the traditional receiver operating characteristic (ROC) curve and its AUC. DL studies are also notable for testing the generalization of their models on entirely new datasets, including data from new areas, new acquisition times, or even new sensors.

**Keywords:** accuracy assessment; thematic mapping; feature extraction; object detection; semantic segmentation; instance segmentation; deep learning

## 1. Introduction

The importance of assessment of the accuracy of remote sensing (RS) thematic classification has been recognized since the early days of remote sensing [1–11]. Congalton [1,2] and Congalton and Green [3] summarized the evolution of traditional RS accuracy assessment best practices. Today, there is a general consensus regarding the importance of unbiased, randomized sampling to support the generation of summary accuracy data, normally presented in the form of a table called the confusion matrix, or error matrix. This table forms the basis for calculating summary metrics, most commonly the overall accuracy (OA), the Kappa statistic (though the use of this statistic has been challenged [12,13]), and the class-specific statistics of user's (UA) and producer's accuracy (PA). Although there are many additional metrics that are sometimes used (e.g., see Pontius and Millones [13]) and the implementations of these methods can vary depending on whether the classification is pixel-based or part of a geographic object-based image analysis (GEOBIA) [8,14–19], the focus on OA, UA, PA, and sometimes Kappa, is very common in traditional RS methods.

Over the last decade, deep learning (DL) methods that rely on convolutional neural networks (CNNs) have increasingly become a central focus in RS classification. CNNs

are a type of artificial neural network (ANN), which are nonparametric, machine learning (ML) methods that use interconnected neurons or nodes organized into layers to predict an output, such as a classification label, from input data, such as image bands [20–27]. DL expands the basic ANN framework by incorporating many hidden layers to allow the modeling of more complex patterns than what would be possible with a small number of hidden layers [20–25]. In CNN classification, the input nodes for the classifier include not just a single pixel, but local groups of adjacent pixels. The CNN learning process incorporates the determination of appropriate convolutional operations and weights associated with multiple kernels or moving windows, allowing the network to model useful spatial context information at multiple spatial scales [20–25]. This process is conceptualized in Figure 1, which represents a subset of feature maps produced when learned kernels were applied to an image chip from the LandCover.ai dataset [28]. Note that spatial context, spectral information, edge, and textural information is highlighted by different filters and different convolutional layers, allowing a high degree of data abstraction. This modeling of spatial context is especially applicable for extracting features from high spatial resolution imagery that often have reduced spectral resolution [29,30].

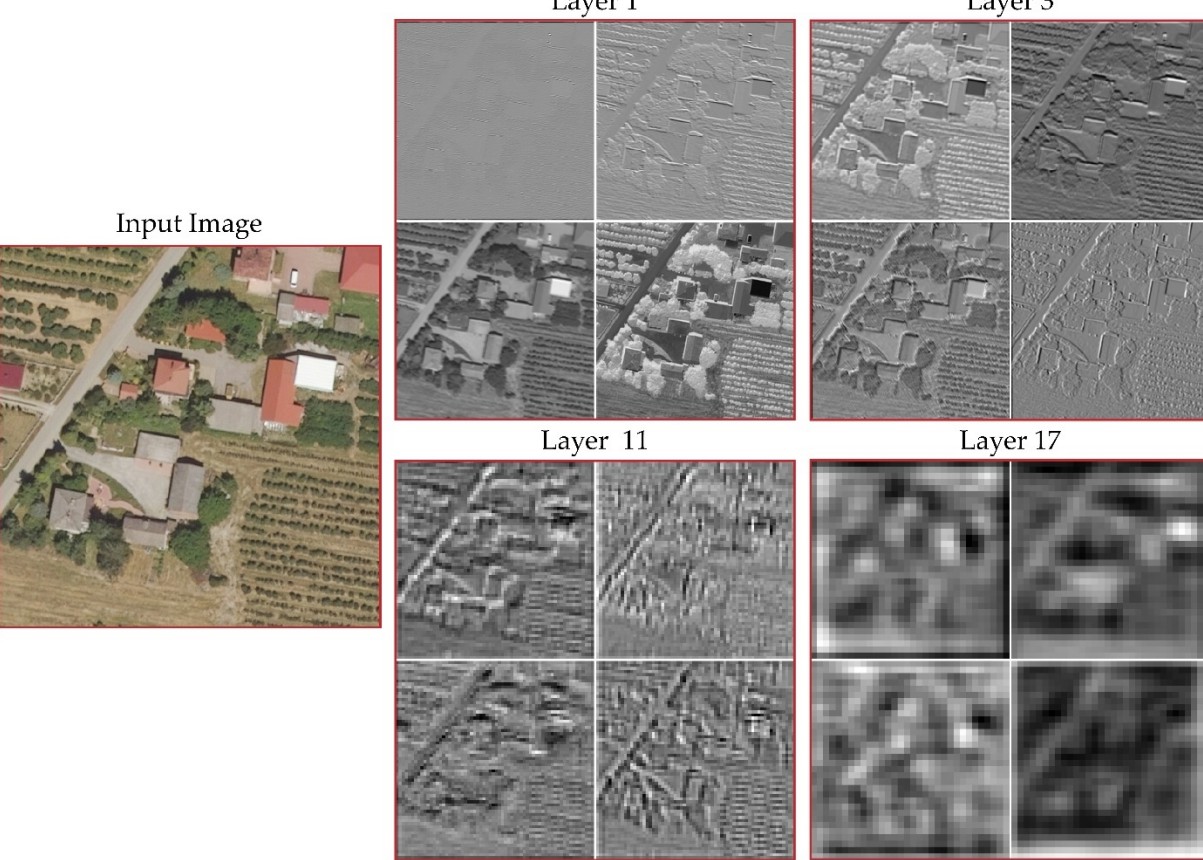

**Figure 1.** Example feature maps generated by convolutional operations. Example data are from the LandCover.ai dataset [28]. The subset of feature maps shown for each convolutional layer are from a ResNet-18 backbone [31] with ImageNet [32] pre-trained weights. Many feature maps are generated by each convolutional layer, so only a subset of four feature maps from four layers are shown here.

DL methods have yielded impressive performance for a wide variety of thematic mapping and feature extraction tasks [24,25,30,33–42], and have proved their effectiveness in operational mapping. For example, Microsoft used DL to generate a building footprint dataset for the United States (US) containing nearly 125 million features [43]. Commercial software environments, such as ArcGIS Pro [34,35], eCognition [36], ENVI [44,45], and

Erdas Imagine [46,47], now offer DL toolsets and modules. DL can also be implemented in open-source environments using a variety of platforms and application programming interfaces (APIs), such as Tensorflow/Keras [48,49] and Torch/PyTorch/fast.ai [50–52]. DL with CNNs may quickly replace the current operational standards for supervised classification, such as random forests (RF) and support vector machines (SVM) [53]; however, the computational intensity and training data requirements of DL methods may slow their adoption.

DL fits in well with the long-held interest in the RS community in incorporating spatial pattern and context in classification, as shown by the many studies that incorporate texture or that employ a GEOBIA approach. Furthermore, key DL applications, such as identifying the class an image belongs to and delineating instances of an object, pre-date DL technology. For example, Aksoy et al. [54] used a Bayesian framework and a visual grammar to classify images. In addition, there is an important subfield of RS involving the delineation and counting of individual trees (e.g., see Warner et al. [55] and Brandtberg et al. [56]). Despite the fact that therefore RS DL is not an inherently new RS application (although the methods are of course new and distinctively different), DL papers do not appear to follow the terminology and metrics of the traditional remote sensing classification literature. Instead, the remote sensing DL community appears to have embraced the terminology and metrics of the artificial intelligence (AI) as well as computer vision communities, from where DL methods generally evolved.

Given the long history of the development of RS accuracy terminology and methods, this switch to an alternative approach is remarkable. This paper, Part 1 of a two-paper sequence, therefore, uses a systematic review of a random sample of the literature from 2020 to summarize RS DL current accuracy assessment approaches. We also explore how RS DL measures relate to traditional RS accuracy measures and investigate whether the RS DL approach and accuracy metrics are fundamentally different, or based on similar approaches with simply new names for measures that traditional RS scientists are familiar with. In our second paper (Part 2; Maxwell et al., in review) we explore the implications of our findings presented here and recommend best practices for assessment of CNN-based products and model generalization.

The rest of this paper is organized as follows. In Section 2, we start with background on the purpose of accuracy assessment and a brief overview of traditional RS accuracy assessment methods. This is followed in Section 3 by an explanation of the literature review methods employed. The results of the literature are summarized in Section 4, focusing on the major accuracy metrics used in RS DL papers. We discuss a range of general issues arising from our review in Section 5, exploring broader themes such as how RS DL studies address class prevalence (the proportion of each class in the landscape) and to what extent DL studies use the population confusion matrix in deriving accuracy metrics. In Section 6, we present our overall conclusions.

## 2. Background

### 2.1. Traditional Remote Sensing Accuracy Evaluation

Traditional RS accuracy evaluation has been extensively described in previous literature. Therefore, we provide only a brief overview of the topic here. Readers interested in more detail on this topic should consult other sources such as Congalton and Green [3], Stehman and Foody [11], and Foody [5].

Stehman and Czaplewski [10] identify three components of RS accuracy assessment: (1) The response design is the choice of sampling unit, for example, point, pixel, or polygon. (2) The sampling design specifies how the samples are selected, for example, using a random or systematic approach. In general, only probability-based sampling allows inference of statistically rigorous map accuracies [5–11,57–67]. (3) The analysis is the protocol that specifies the accuracy measures and how population-based estimates of those metrics are calculated from the sampling results [10,67].

### 2.2. The Purpose of Accuracy Assessment

The calculation of accuracy metrics provides three main practical benefits, which, though overlapping, are sometimes in tension with one another. First, accuracy metrics allow the benchmarking and comparison of methods. By using the terminology of AI studies, RS DL studies facilitate the comparison of their results with the broader computer science community, though, of course, at the potential expense of communication with the RS community. For studies that wish to contribute to AI open challenges, or use standard AI benchmark datasets, use of specified AI accuracy protocols and terms may be useful or even required. For example, a number of RS DL studies draw upon the Microsoft Common Objects in Context (COCO) project (cocodataset.org), which provides its own evaluation code and associated terminology [43]. In comparing methods, there is often an implicit desire to identify a "best" method, or, at least, to rank the methods. It is clearly simpler to do so with a single metric, rather than with multiple metrics, which might give conflicting results. The challenge is that accuracy has multiple components, and generally it is possible to generate the same summary value from different combinations of values of the accuracy components. This suggests that if summary metrics are presented, the underlying data (i.e., the confusion matrix) should also be available to the reader. However, not all studies seek a single metric for comparing algorithms; the COCO project notably specifies the use of a wide range of metrics when using their accuracy assessment approach [43].

A second major benefit offered by accuracy metrics is insight into an algorithm's performance, and, in particular, its strengths and weaknesses. This may require multiple accuracy metrics. Although, for this purpose, the relationship of these metrics to those in the literature may not be important, the use of metrics with an intuitive or well-known meaning will facilitate interpretation and communication of the results [68]. A final major benefit of accuracy assessment is to give insight into the real-world application of the method studied. For predicting real-world performance, the testing sample needs to reflect the population characteristics of the data which the algorithm will likely use [69,70].

### 2.2.1. Deep Learning Accuracy Assessment Example Use Cases

Other than practically assessing derived thematic products, appropriate, consistent, rigorous, and well-documented accuracy assessment methods are key for benchmarking and quantifying improvements resulting from augmentation of existing and development of new DL methods, as mentioned above. For example, Zhou et al. [71] introduced Unet++ and quantified improved classification performance in comparison to UNet and Wide-UNet. More recently, Sun et al. [72] compared their proposed Circle-UNet architecture with a variety of existing UNet architectures to document improved semantic segmentation performance. Assessment methods are also necessary for quantifying model sensitivity to hyperparameter settings (e.g., Li and Hsu [73]), such as architecture augmentations; loss metric, optimization algorithm, or learning rate used, and learning rate scheduling applied. Accuracy assessment is also important when studying the impact of training data quantity (i.e., ablation studies), quality, and/or augmentation (e.g., [38,68,74,75]).

Further, accuracy assessment is central to a wide variety of DL RS studies, not just algorithm comparison and development. Assessment is necessary for exploring the impact of feature space, feature reduction or selection, sensor calibration, and input image selection (e.g., [76–78]). As specific examples, Yang et al. [78] compared their proposed hyperspectral band selection method with existing techniques to document improved classification performance while Abdalla et al. [77] assessed their combined DL and *k*-means method for color calibration. Witharana et al. [79] explored the impact of different data fusion and pansharpening methods on subsequent DL classification performance.

DL techniques have been shown to be especially applicable for extracting information from high spatial or spectral resolution datasets [76–78,80–84]; however, key issues must be investigated, which require appropriate accuracy assessment protocols. Zang et al. [80] noted the complexity of land use mapping from high spatial resolution datasets resulting from variability and inconsistency between images and illuminating conditions, and high

levels of detail and variability in class representations. Additionally, the use of transfer learning (e.g., Li et al. [29]) and/or the management of network size and number of trainable model parameters often require a subset of multispectral or hyperspectral bands be selected, as simply using visible spectrum bands may not be optimal due to loss of information content [78,83,84]. For example, Bhuiyan et al. [83] documented that the selected three band subset of WorldView-2 multispectral imagery used significantly impacted the DL classification accuracy for permafrost tundra landform mapping. Similarly, Cai et al. [84] quantified the importance of different input variable combinations for semantic segmentation of point cloud data. Highlighting the importance of such issues, methods have been developed to select appropriate bands from hyperspectral imagery to support DL-based classification (e.g., Yang et al. [78]), calibrate imagery for consistency (e.g., Abdalla et al. [77]), and select images with minimal cloud cover (e.g., Park et al. [76]).

Accuracy assessment is also important for exploring model generalization and transferability to new data and/or geographic extents. For example, Maggiori et al. [85] assessed building detection models when extrapolated to new cities while Maxwell et al. [38] assessed the extraction of historic surface mining extents when applied to new topographic maps in new regions. Robinson et al. [86] explored the extrapolation of general land cover models trained in the eastern United States to new regions of the country.

As this section highlights, accuracy assessment is a required component of a wide variety of RS DL studies, which highlights the need for and adherence to appropriate and rigorous accuracy assessment methods to advance the field and minimize misleading or incorrect findings. Given the ubiquitous use and requirement for accuracy assessment in RS studies and the further complexity of rapid advancement in methods, such as those relying on DL, we argue that it is of great importance to evaluate current assessment protocols and make recommendations for best practices moving forward.

### 2.2.2. The Confusion Matrix

An RS map accuracy assessment normally involves the spatial overlay of the reference samples on the classified map. The number of samples in each combination of reference class and predicted class is then summarized in the sample confusion matrix. Generally, the columns represent the reference classification label and the rows represent the classification label, though this convention is not always followed.

A key distinction that is sometimes lost is that this sample confusion matrix does not necessarily estimate the properties of the map, unless the samples were collected with a purely random sampling design [7,8,10,11,58,59,62–66,87,88]. Only a pure random sample (or in some cases, a complete census) provides a direct estimate of the population proportions in the landscape for the confusion matrix. For all other sampling designs, such as stratified random sampling, the values in the sample confusion matrix do not reflect landscape proportions [10,11,62–64,66,88]. Thus, an important step in designing the accuracy analysis protocol is to specify how the population confusion matrix is estimated from the sample matrix. For example, Stehman and Foody [11] describe the procedure if the classification itself is used to stratify the sampling. Stehman [66] describes the procedure when samples derived from a stratification of one classified map are applied to another map, a situation that may occur in studies comparing multiple classifications.

An appropriately configured population confusion matrix has entries that reflect the proportion of each tabulated category in the landscape (Table 1). Sometimes, studies report a confusion matrix that has been "normalized", in which the class proportions have been iteratively reconfigured such that all row and column totals are equal [3,11,87]. However, such a normalized matrix and the derived accuracy measures will not represent the actual map properties, but instead, a hypothetical situation where the classes have equal prevalence. Since classes that comprise a small proportion of the landscape have low prior probabilities, they are generally more difficult to map than classes that are more common, if all else is equal. Therefore, retaining the class prevalence is important for calculating

the appropriate accuracy measures. For this reason, normalization of the confusion matrix should be avoided [8,87].

**Table 1.** The population confusion matrix. A, B, and C represent the class labels. $P_{ij}$ represents the proportion of the map area that is classified as class $i$ and is class $j$ in the reference data. The + symbol is used to represent summation, with the summation over a column for the + symbol in the first subscript position, and across the row in the second subscript position. UA = user's accuracy and PA = producer's accuracy.

| | | Reference | | | Row Total | UA |
|---|---|---|---|---|---|---|
| | | **A** | **B** | **C** | | |
| Classification | A | $P_{AA}$ | $P_{AB}$ | $P_{AC}$ | $P_{A+}$ | $P_{AA}/P_{A+}$ |
| | B | $P_{BA}$ | $P_{BB}$ | $P_{BC}$ | $P_{B+}$ | $P_{BB}/P_{B+}$ |
| | C | $P_{CA}$ | $P_{CB}$ | $P_{CC}$ | $P_{C+}$ | $P_{CC}/P_{C+}$ |
| | Column total | $P_{+A}$ | $P_{+B}$ | $P_{+C}$ | | |
| | PA | $P_{AA}/P_{+A}$ | $P_{BB}/P_{+B}$ | $P_{CC}/P_{+C}$ | | |

### 2.2.3. Summary Metrics Derived from Confusion Matrix

Once the population confusion matrix has been estimated, a variety of summary metrics can be derived (Table 2). The overall accuracy is the proportion of the map correctly classified, divided by the area of the entire map. Many RS studies also report the Kappa statistic as a measure of the success of classification compared to what could be achieved by chance agreement [3,8,60]. However, the necessity for applying a correction for chance agreement and the appropriateness of Kappa as an estimate of chance agreement have been criticized, resulting in strong recommendations to discontinue the use of Kappa [8,12,13].

**Table 2.** Multiclass metrics derived from the confusion matrix.

| Measure | Type of Measure | Equation |
|---|---|---|
| Overall Accuracy (OA) | Integrated summary | $\frac{\text{Area of map correctly labeled}}{\text{Total area of map}}$ |
| Kappa | Integrated summary | $\frac{(\text{OA}-\text{expected agreement})}{(1-\text{expected agreement})}$ |
| User's Accuracy (UA) | Class-based | $\frac{\text{Area of map correctly labeled as class } x}{\text{Area of predicted map labeled class } x}$ |
| Producer's Accuracy (PA) | Class-based | $\frac{\text{Area of map correctly labeled as class } x}{\text{Area of reference map labeled class } x}$ |

The most common class-based accuracies are user's accuracy (UA) and producer's accuracy (PA) [3,8]. UA and PA are ratios representing the proportion of correctly classified pixels for a specific class relative to the pixels classified as that class by the algorithm for UA, or labeled that class in the reference data, for PA (Table 2). It is generally necessary to report both user's and producer's accuracies, since it is possible to have a high user's accuracy with a low producer's accuracy, or vice versa.

### 3. Literature Review Methods for Surveying Accuracy Evaluation of CNNs in Remote Sensing

We sampled a random selection of 100 papers recently published in RS journals as the basis of our summary review of how RS DL papers approach accuracy assessment. We used Clarivate's Web of Science database and limited the publication year to 2020. As our focus was on DL in the RS community, we limited our search to the following eight RS journals, which a preliminary search indicated were the primary outlets for the majority of RS DL papers: *IEEE Geoscience and Remote Sensing Letters, IEEE Journal of Selected Topics in Applied Earth Observations and Remote Sensing, IEEE Transactions on Geoscience and Remote*

*Sensing, International Journal of Remote Sensing, ISPRS Journal of Photogrammetry and Remote Sensing, Remote Sensing, Remote Sensing Letters*, and *Remote Sensing of Environment*.

After further experimentation, the following search query was used:

AB = (((CONVOLUTION* NEURAL NETWORK*) OR (CNN) OR (DEEP LEARNING))
AND ((SCENE CLASSIFICATION) OR (SCENE LABEL*) OR (OBJECT DETECTION)
OR (SEMANTIC SEGMENTATION) OR (INSTANCE SEGMENTATION)))

The literature search focused on terms used in the abstract, as indicated by the AB in the above query. The * represents a wildcard, to ensure that variations of terms, such as both NETWORK and NETWORKS, would be flagged. The keywords scene classification, scene label*, object detection, semantic segmentation, and instance segmentation relate to the four major applications of CNNs described below and summarized in Figure 2.

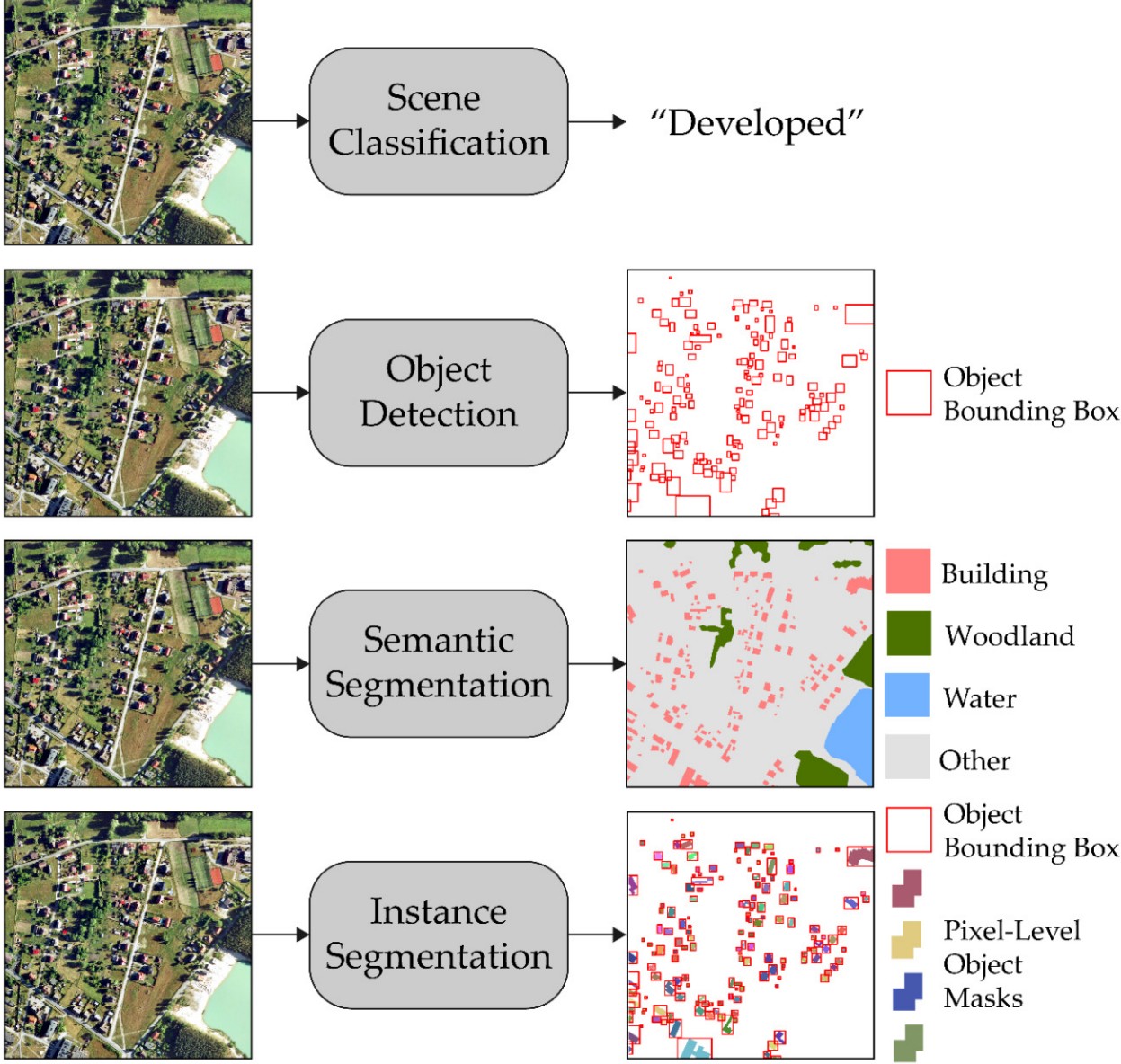

**Figure 2.** The four main types of thematic mapping and object detection using CNNs applied to remotely sensed data. Example data are from the LandCover.ai dataset [28].

- Scene classification, sometimes referred to as scene labeling, involves classifying an entire image or image chip to a single category, or multiple categories, with no localized detection of features within the image. For example, an entire scene could be recognized as an example of a developed area or a developed area and woodlands [20,21,34,35].
- Object detection includes the estimation of the location of occurrence of features within the image extent. The output is a bounding box for each detected feature, along with an associated class label and probability [20,21,24,25,89,90].
- Semantic segmentation is similar to traditional, pixel-based classification. Here, each pixel is assigned to a thematic class. It is also possible to obtain the probability of each pixel's membership in each class [71,91–96].
- Instance segmentation differentiates and maps the boundaries of each unique occurrence of the classes of interest. For example, each building in an image could be detected as a separate instance of the "building" class. The outputs include bounding boxes for each instance, class probabilities, and pixel-level feature masks [24,25,97].

The Web of Science search resulted in 246 papers. A preliminary screening found eight of these papers did not have RS DL classification as a primary focus and were therefore deleted from the list. From the remaining 238 papers, a random subset of 100 papers was generated to form the focus of the review. Three of the 100 papers each comprised two separate studies, and thus, the total number of studies evaluated was 103 (Table 3). Of these 103 studies, just under half were binary classifications or identified a single class (for simplicity, we subsequently refer to both of these as binary classifications), the remainder were multiclass classifications. Semantic segmentation was the most common CNN classification type, and instance segmentation the least common. Scene classification is generally conceptualized as a multiclass problem, and thus, no examples of binary scene classification were identified.

**Table 3.** Studies surveyed in the literature review by CNN classification type. Of the 100 papers that were surveyed, three papers each had two studies of different types, resulting in a total of 103 studies surveyed.

| CNN Classification Type | Number of Studies | | |
| --- | --- | --- | --- |
| | Binary and Single Class Classifications | Multiclass Classifications | Total |
| Scene Classification | 0 | 12 | 12 |
| Object Detection | 18 | 13 | 32 |
| Semantic Segmentation | 20 | 33 | 52 |
| Instance Segmentation | 3 | 4 | 7 |
| Total | 41 | 62 | 103 |

Each of the 103 studies was reviewed to determine the accuracy measures reported. We focused exclusively on accuracy measures reported regarding the classifier performance using independent test data, and excluded any measures used only as part of training or optimizing the classifier.

## 4. Accuracy Metrics Commonly Used in Remote Sensing CNN Classifications

### 4.1. The Confusion Matrix in RS CNN Studies

4.1.1. The Binary Confusion Matrix: True and False Positives and Negatives

The terminology generally used in the accuracy evaluations of RS CNN classifications has its origins in the binary confusion matrix, with the class of interest referred to as the positive case, and the background as the negative case (Table 4). The binary confusion matrix has four entries: the number of true positive (TP) and true negative (TN) samples, which are respectively those that are correctly mapped as positive and negative, and the two error categories of false positive (FP) and false negative (FN) samples, which represent

the number of negatives incorrectly mapped as positives, and vice versa (Table 4). In statistical hypothesis testing, FPs are referred to as Type I errors, and FNs as Type II errors. Due to the range of CNN classification types, the numbers of TP, FP, TN, and FN potentially represent pixels, objects, or scenes. For objects, TN is commonly not defined. Therefore, for object detection and instance segmentation, a full confusion matrix may only comprise the remaining three components.

**Table 4.** Conceptualization of a binary classification confusion matrix. TP = True Positive, TN = True Negative, FP = False Positive, and FN = False Negative.

|  |  | Reference Data | |
|---|---|---|---|
|  |  | **Positive** | **Negative** |
| Classification Result | Positive | TP | FP |
|  | Negative | FN | TN |

Although the binary confusion matrix is an important concept for deriving other accuracy measures, none of the binary classification papers surveyed reported a complete binary confusion matrix (Figure 3), though three other papers reported a subset of the matrix, for example, just FP or the combination of FP and FN.

### 4.1.2. The Multiclass DL CNN Confusion Matrix

For multiclass classification, even if a full confusion matrix is presented in the paper, the terminology of TP, FP, FN, and FP is often used in describing the classification results. The main difference is that in multiclass classification, these terms are all class-specific, and other than TP, represent the sum of multiple cells in the complete multiclass confusion matrix illustrated in Table 1. For example, FP is the sum of the row representing the proportion of samples labeled to a particular class by the classifier, minus the TP proportion for that class. When presented in a paper, these confusion matrices are often color-coded, to make it easier to discern the high and low values in the table. Unlike the situation for DL binary classifications, DL multiclass classifications do sometimes report complete confusion matrices. For example, 42% of scene classification and 30% of semantic segmentation studies included an entire confusion matrix (Figure 3).

### 4.1.3. The Sample vs. the Population Confusion Matrix

Almost all the papers surveyed incorporated a sampling strategy that divide training and evaluation samples using a method such as random sampling (49% of studies), stratified random sampling (3%), systematic sampling (1%) or by locating evaluation samples in an entirely different area (34%). The remaining 13% used either a purposive approach in dividing training and testing samples, or the method was unclear. However, in almost half of the studies, the original selection of the samples can be attributed to a purposive selection (48%), or the method used is not clear (11%). Only 41% of studies appear to use a sampling strategy that is designed to allow the eventual estimation of the population confusion matrix, i.e., are a probability-based sample. It is notable that community benchmark datasets often only label a purposive set of samples, thus also, not following a probability-based sampling approach.

Of the 17 studies that reported a complete confusion matrix, only 4 present numbers that appear to be equivalent to the landscape proportions of those classes, and thus meet the definition of a population error matrix (Table 5). In the rest of the studies (13), the numbers in the confusion matrix appear to have been normalized by dividing by the total number of samples in the reference category (11 studies) or the predicted category (2 studies), so that the columns or rows sum to 100%. The rationale for this normalization is generally not explained, but is apparently designed to force each class to have an equal weighting in the matrix, as with the now discouraged traditional RS approach of a "normalized" confusion matrix.

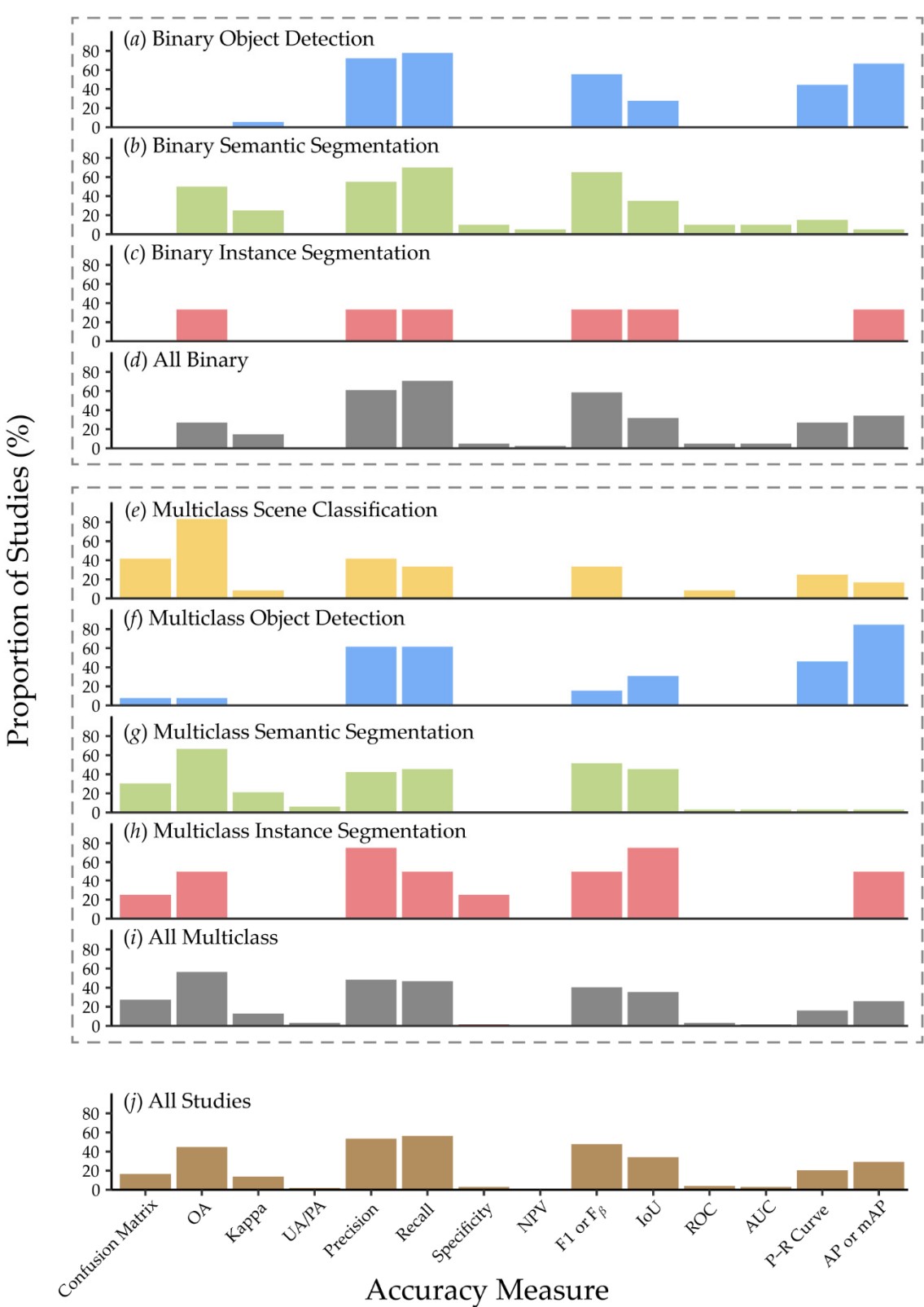

**Figure 3.** Proportion of DL studies reporting various accuracy measures, grouped by type of classification (binary vs. multiclass) and type of CNN algorithm (scene classification, object identification, semantic segmentation, and instance segmentation). OA = Overall Accuracy, UA/PA = User's Accuracy or Producer's Accuracy, NPV = Negative Predictive Value, F1 or $F_\beta$ = any form of F1 or $F_\beta$, IoU = Intersection-over-Union, ROC = Receiver Operating Characteristic Curve, AUC = Area Under the ROC Curve, P-R Curve = Precision-Recall Curve, AP or mAP = Average Precision or mean Average Precision.

**Table 5.** Type of confusion matrix presented in the surveyed literature.

| Type of Confusion Matrix | Number of Studies |
|---|---|
| Reference category values add to 100% | 11 |
| Predicted category values add to 100% | 2 |
| Values in entire table add to 100% (or values represent landscape proportions) | 4 |
| Total | 17 |

### 4.2. Accuracy Metrics Derived from the Confusion Matrix and Commonly used in CNN Applications

Table 6 presents an overview of the main metrics commonly reported in RS DL studies. The table shows the most common name in the first column, and other names found in our literature review for the same accuracy measure in the second column. The large number of entries in this second column indicates that many common DL accuracy assessment metrics are known by multiple names. Most of the accuracy measures in the table are designed to scale from 0 to 1.0, with 0 representing total disagreement between the reference and classification data, and 1.0 representing total agreement. However, Kappa and the Matthews correlation coefficient are scaled from −1.0 to +1.0 [12,98]. Tharwat [98] provides an excellent overview of the major DL accuracy measures and the variety of names associated with them in the DL literature.

**Table 6.** Classification accuracy assessment metrics most commonly used in the DL literature, with the associated equation for a binary classification matrix. Citations are from the literature review for illustrative purposes, and are not meant to be comprehensive.

| Generally Accepted or Most Commonly Used Name of Measure in RS DL | Other Names Used in the RS DL Literature | Equation | Relation to Traditional RS Measures |
|---|---|---|---|
| Overall Accuracy (OA) [99] | Percent Correct Classification [100] Pixel Accuracy [101] | $\frac{TP+TN}{TP+TN+FP+FN}$ | Overall Accuracy |
| Recall [102] | Sensitivity [101] True Positive Rate (TPR) [101] Overall Accuracy [103] Detection Probability [68] Hit Rate [104] | $\frac{TP}{TP+FN}$ | PA for positives |
| Precision [102] | Positive Predictive Value (PPV) [101] | $\frac{TP}{TP+FP}$ | UA for positives |
| Specificity [105] | True Negative Rate (TNR) [101] | $\frac{TN}{TN+FP}$ | PA for negatives |
| Negative Predictive Value (NPV) [101] | | $\frac{TN}{TN+FN}$ | UA for negatives |
| False Positive Rate (FPR) [106] | Probability of False Detection [107] False Alarm Probability [100] | $\frac{FP}{TN+FP}$ | 1− (PA for negatives) |
| False Negative Rate (FNR) [1] | Missing Detection Probability [100] Missing Alarm [108] Misidentification Score [109] | $\frac{FN}{TP+FN}$ | 1− (PA for positives) |
| False Discovery Rate (FDR) [1] | False Alarm Probability [68] Commission Error [110] | $\frac{FP}{TP+FP}$ | 1− (UA for positives) |
| Balanced Accuracy [101] | | $\frac{1}{2}$(Recall + Specificity) | |
| Matthews Correlation Coefficient (MCC) [101] | | $\frac{(TP \times TN)-(FP \times FN)}{\sqrt{(TP+FP)\times(TP+FN)\times(TN+FP)\times(TN+FN)}}$ | |
| F1 [111] | F-measure [112] (Liu et al. 2020) F-score [113] F$_\beta$ Score [114] Sørensen–Dice Coefficient [115] | $\frac{2\times\ Precision\ \times\ Recall}{Precision\ +\ Recall}$ or $\frac{2\times\ TP}{2\times TP+FN+FP}$ | |
| Intersection-over-Union (IoU) [99] | Jaccard Index [115] | $\frac{TP}{TP+FP+FN}$ | |

[1] FNR and FDR are not commonly used in RS DL accuracy assessments. However, for consistency with the use of these terms elsewhere in the surveyed literature (e.g., the work of [116] uses FNR), and with Tharwat [98], we list them here.

Figure 3 above summarizes the frequency at which each accuracy measure is used by papers that focus on binary and multiclass classification types, as well as by scene classification, object detection, semantic segmentation, and instance segmentation applications. A comparison of the graphs indicates that some measures (for example, precision and recall) are used for all types of classification applications, although it is notable that no single measure is used by every single study, even within one category of applications (e.g., multiclass scene identification). In other words, there is no single universally accepted accuracy measure. On the other hand, some measures tend to be associated with specific classification types and applications. For example, PR-curves are most strongly associated with binary and multiclass object detection, as well as multiclass scene classification.

### 4.2.1. Overall Accuracy and Kappa

Overall accuracy (OA) and Kappa are notably the two measures reported in RS DL studies that are usually identical in name and definition to those used in traditional remote sensing studies. However, as Table 6 demonstrates, OA, like most RS DL accuracy measures, is sometimes given alternative names such as percent correct classification, pixel accuracy, or accuracy. OA is reported in under half of RS DL studies (46%, Figure 3). Part of the reason for the low number of studies reporting OA is that this metric is not a good fit for object identification or instance segmentation, since the TN category is normally not defined for objects. Another reason suggested for either not reporting or de-emphasizing OA is that rare classes are given only a low weighting in the OA calculations.

Kappa is reported only occasionally in RS DL studies (14%), most commonly in semantic segmentation applications, where it is reported in 25% of binary studies and 21% of multiclass studies.

### 4.2.2. Recall and Precision

Recall and precision, the most common summary metrics reported in the RS DL literature, are reported respectively in 71% and 61% of binary studies, and in 50% and 48% of multiclass studies. Table 6 shows these metrics are equivalent to the traditional RS measures of PA and UA for the positive class.

### 4.2.3. Specificity and Negative Predictive Value

Specificity and negative predictive value (NPV) are measures of the negative class accuracy and are only occasionally reported in DL RS studies, respectively 3% and 1% of studies. They are equivalent to the traditional RS accuracy metrics of PA and UA for the negative class. The specificity metric is most often reported as a component of the receiver operating characteristic (ROC) curve, discussed in more detail in Section 4.3.1.

### 4.2.4. False Positive Rate, False Negative Rate, False Discovery Rate

The three measures of false positive rate, false negative rate, and false discovery rate represent 1.0 minus the value of specificity, recall, and precision, respectively. Since each of the latter three measures is perfectly correlated with each of the former three, it is redundant to report both corresponding pairs of measures. These measures were generally only occasionally reported in the survey (<5% of studies).

### 4.2.5. Balanced Accuracy and Matthews Correlation Coefficient

A range of metrics combine previous metrics or the elements of the confusion matrix in additional ways. Balanced accuracy, for example, is simply the average of recall and specificity. The Matthews correlation coefficient represents the correlation between the reference and predicted classifications [98]. Both these measures are only occasionally reported.

### 4.2.6. F1

Many authors report the combined class-based measure F1, also called F1 score, F-score, Sørensen–Dice Coefficient, or Dice Coefficient. This measure is often described as

the harmonic mean of recall and precision [98], which are themselves popular accuracy measures. However, as also shown by Table 6, it can also be useful to understand F1 in terms of TP, FN, and FP in order to compare it more directly to other accuracy metrics. The F1 statistic is reported in 59% of binary studies, and as a class-specific measure in 40% of multiclass studies.

The F1 statistic is a specific version of the general $F_\beta$ score, which is occasionally reported in DL literature.

$$F_\beta \text{ Score } = \left(1+\beta^2\right) \frac{\text{Precision } \times \text{ Recall}}{\beta^2 \times \text{ Precision } + \text{ Recall}} \tag{1}$$

$$= \frac{\left(1+\beta^2\right)(\text{TP})}{(1+\beta^2)(\text{TP}) + \beta^2(\text{FN}) + (\text{FP})} \tag{2}$$

As shown by Equation (2), $\beta$ represents the weighting applied to TP and FN relative to FP. When $\beta = 1.0$ (i.e., the F1 measure), FN and FP are equally weighted; higher values of $\beta$ exponentially increase the weighting of both TP and FN relative to FP. Other values used for $\beta$ in the surveyed papers include 0.3 [117] and 2 [114]. Adding further complexity to the F1 score is that some studies use alternative, simpler versions for the equation for calculating F1, e.g., (Precision × Recall)/(Precision + Recall) [40].

### 4.2.7. Intersection-Over-Union (IoU)

Intersection-over-union (IoU), or the Jaccard index, is a ratio of the intersection of the reference and classified samples with the union of the two groups. As Table 6 indicates, IoU has an equation similar to that of the F1 measure, making the two measures correlated, though not linearly because F1 has twice the weighting of the intersection (TP) in the ratio. As a result, other than at the extremes of 0.0 and 1.0, F1 values are higher than IoU for the same map (Figure 4).

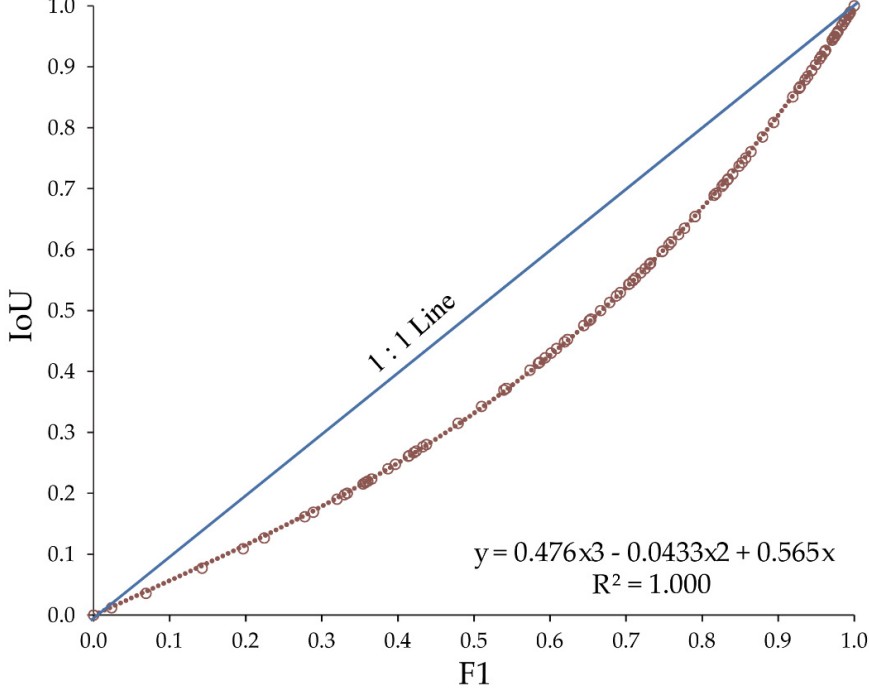

**Figure 4.** Graph of F1 vs. IoU for 100 randomly generated points, illustrating the correlated nature of these two measures of accuracy. Each random reference point is represented as a circle with the associated dotted line representing the perfect and non-linear correlation between F1 score and IoU. The blue line represents a perfectly linear, 1:1 correlation for reference.

The IoU ratio can be calculated based on pixels or bounding boxes that encompass individual objects in the image (Figure 5). The bounding box concept has its origins in the computer vision community, and IoU is commonly used in object identification and instance segmentation (Figure 3). IoU is often used as the threshold that determines if objects are regarded as TP or FN, with 0.5 as the most commonly chosen threshold (e.g., Zhang et al. [40]). In semantic segmentation, IoU is usually applied directly to pixels over the classified image as a whole.

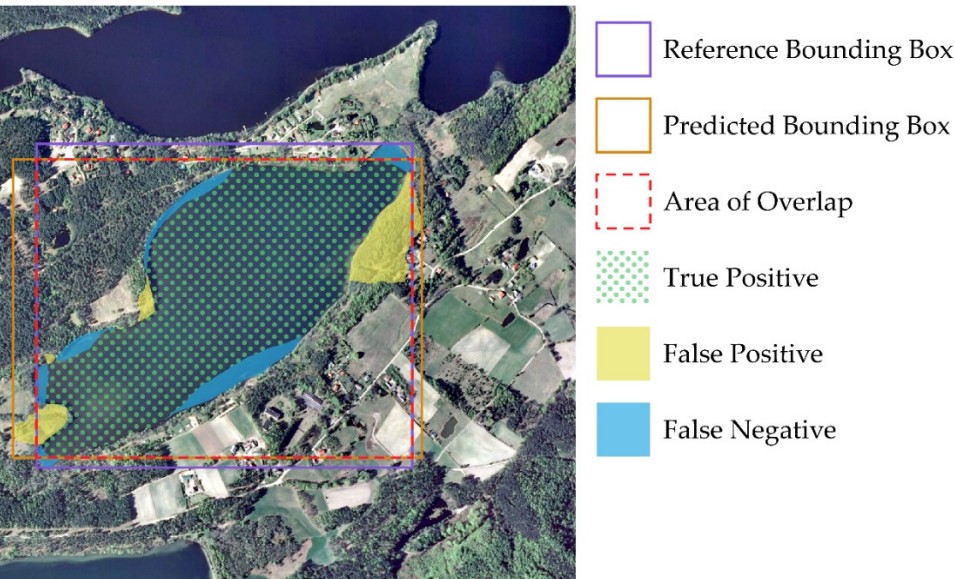

**Figure 5.** Example of intersection-over-union (IoU) from the LandCover.ai dataset [28]. Box-based IoU is shown using the series of rectangles. The equivalent pixel-based IoU is illustrated with the stippled texture as well as blue and yellow colors.

### 4.2.8. Combined Multiclass Metrics

A common approach in RS DL studies is to average individual metrics over all the mapped classes. Examples of such metrics include mean recall (sometimes given the term mean accuracy [118], which for binary data is equivalent to balanced accuracy [101]); mean precision [111,119] (not to be confused with average precision, see Section 4.3.2.); mean F1 [120], and mean IoU (mIOU) [121]. Mean F1, for example, is reported by 15% of multiclass studies. As F1 combines precision and recall, some studies regard mean F1 as a type of overall accuracy, and a replacement for the OA metric (e.g., [33]).

Most of these combined multiclass metrics apply a simple average, with each class weighted equally. However, because such a combined metric implies a map with equal-class prevalence, a number of studies report so-called frequency-weighted (FW) multiclass versions of these metrics, for example, the FW IoU of Singh et al. [122].

### 4.3. Metrics for CNN Classifications with Variable Decision Thresholds

In the description of accuracy measures so far, the decision boundary, or threshold, that discriminates between classes has been conceptualized as having a single, fixed value. However, generally, classes are not 100% separable, and any particular decision threshold requires a tradeoff between minimizing FN and FP. This is illustrated in Figure 6, where the curves represent hypothetical distributions of probabilities of the negative and positive classes, as approximated by kernel density functions applied to the outcome of the classifier. Although the peak probability for the positive and negative classes are clearly different, there is overlap between the categories. The classifier performance can be characterized by setting multiple decision thresholds, allowing the associated number of TP, TN, FP, and FN samples for each threshold to be tabulated [98,123–129]. Two common types of graphs that

explore the accuracy tradeoffs by systematically testing a range of thresholds are discussed in the next two subsections.

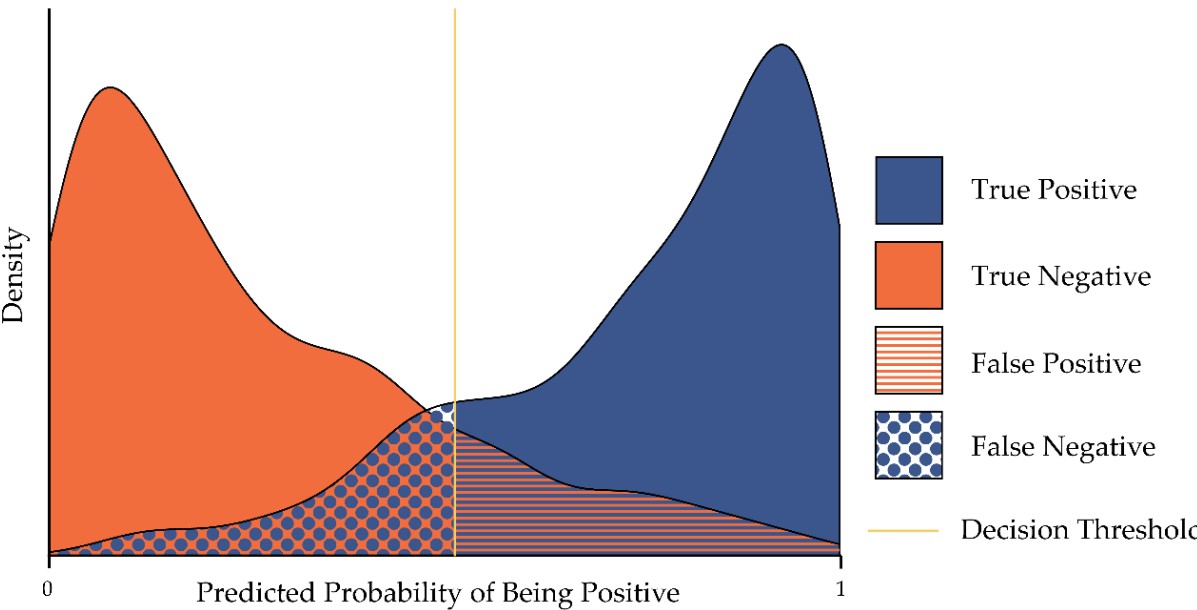

**Figure 6.** Conceptualization of TP, TN, FP, and FN results relative to class probabilities and a defined threshold.

### 4.3.1. Receiver Operating Characteristic Curve (ROC) and Area under the Curve (AUC)

One way of exploring a classifier performance as a function of the decision threshold is the receiver operating characteristic (ROC) curve. An ROC curve plots 1—specificity versus sensitivity (another name for recall, see Table 6), as the decision threshold is varied (Figure 7a). A classifier that is equivalent to guessing, and thus offers no useful information, would plot on the graph diagonal, sometimes termed the baseline, indicated by the dotted line in Figure 7a. An ideal classifier would have an ROC plot represented by two perpendicular straight lines that intersect at the top left corner of the graph, which has coordinates of (0,1), and is equivalent to 100% specificity and 100% sensitivity. Most real classifiers plot in between the ideal and the baseline.

The ROC graph is often summarized by calculating the area under the curve (AUC). The AUC ROC is equivalent to the probability that the classifier will rank a randomly chosen positive (true) record higher than a randomly chosen negative (false) record [98,123–125]. An ideal classifier has an AUC ROC of 1.0 and a classifier that is no better than guessing, i.e., the diagonal line, has a value of 0.5. In the multiclass case, some authors produce multiple ROC curves and associated AUC ROC values by sequentially comparing each single class to all other classes [127]. In our survey of RS DL studies, ROC plots and AUC ROC statistics were occasionally reported (10% or less) for both binary and multiclass semantic segmentations and multiclass scene classification.

The ROC curve and the associated AUC ROC metric's reliance on specificity and sensitivity has been criticized as misleading in cases where the class prevalence is not equal [128], which is probably the norm in many RS applications. Using traditional remote sensing terminology, the criticism is that the ROC curve uses only PA, and ignores UA. A classifier with high PAs does not necessarily have high UAs, especially in the case of imbalanced classes.

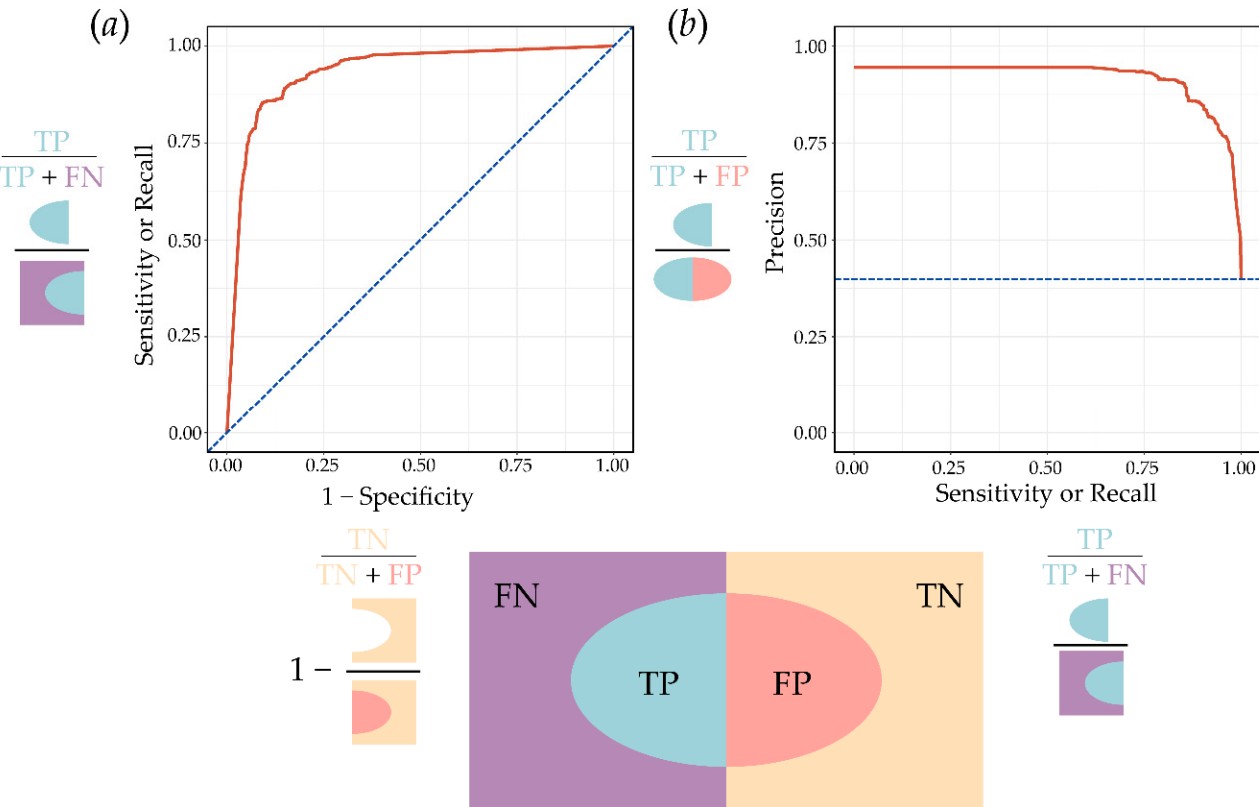

**Figure 7.** Methods for evaluating classifier performance over a range of thresholds: (**a**) ROC curve for a classification with an AUC ROC of 0.93; (**b**) P-R curve.

### 4.3.2. Precision-Recall Curve (P-R Curve), Average Precision, and Mean Average Precision

An alternative to the ROC curve is the precision-recall (P-R) curve (Figure 7b), which plots precision versus recall. Unlike the ROC, the P-R curve's baseline, equivalent to guessing, is the point on the precision axis that has a value equal to the class prevalence. For example, in Figure 7b, the positive class has a roughly 40% prevalence, and therefore, when the decision threshold is set at the extreme level that classifies every unknown as the positive class, the precision will be equal to the prevalence value and the recall will be 100%. By convention, this baseline point is drawn as a horizontal line across the graph, to document that this is the minimum precision, and to serve as a baseline.

As with the ROC curve, it is possible to generate a summary area under the curve (AUC PR) metric [98,129,130]. In the DL community, however, this AUC PR metric is generally referred to as average precision (AP) or mean average precision (mAP). Figure 8 illustrates the method for calculating AP and mAP for classification involving objects. For each individual object, whether that object is a correct or incorrect prediction is defined based on a minimum IoU. For example, for an IoU threshold of 0.6, objects that meet this criterion are labeled as TP, and objects below the 0.6 threshold are labeled as FN (Figure 8a,b). For each IoU threshold, a P-R curve is generated (Figure 8c) with an associated area under the curve, or AP value (Figure 8d) [98,129,130]. AP is then averaged over multiple IoU thresholds (Figure 8e) (e.g., from 0.55 to 0.90 with steps of 0.05, as demonstrated in Figure 8b).

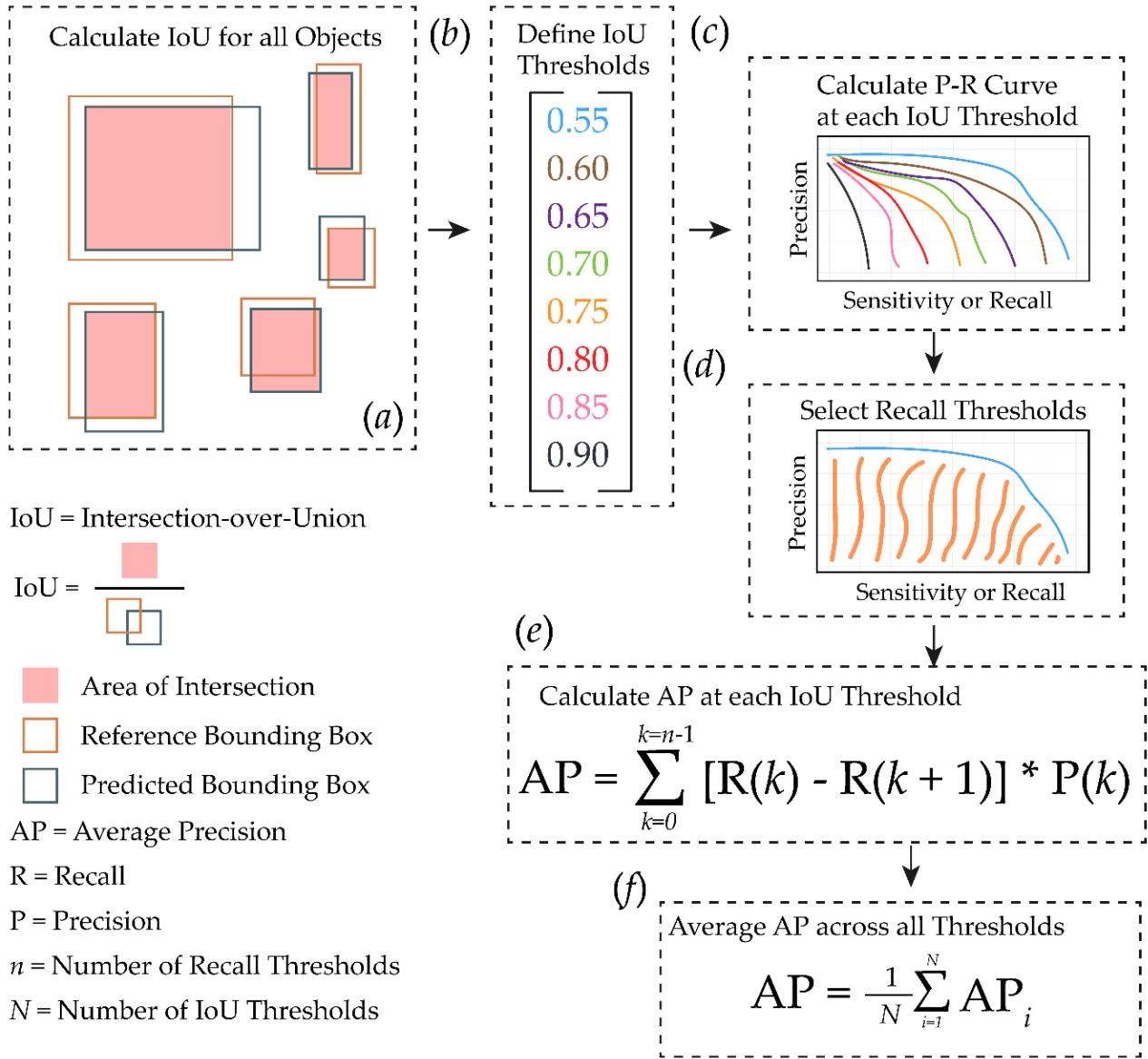

**Figure 8.** Conceptual explanation of average precision (AP) calculated using reference and predicted bounding boxes and eight IoU thresholds: (**a**) IoU for multiple map features; (**b**) Defined IoU thresholds at which to create P-R curves; (**c**) Multiple P-R curves, one for each defined IoU threshold; (**d**) Calculate area under each P-R curve; (**e**) Average AP calculated across all IoU thresholds; (**f**) Average AP across all thresholds.

There is, however, some inconsistency in the literature in the use of the terms AP and mAP. Some authors use AP in reference to a single class and mAP for the average over multiple classes (e.g., [40,89]), yet others refer to AP calculated for a single class at a single IoU threshold as mAP [131]. Following trends in the computer science DL community [132], many recent papers no longer differentiate between AP and mAP, and use AP as a general term (e.g., [133]). The P-R curve was reported by 20% of the studies surveyed and AP and/or mAP by 29% of the studies surveyed.

*4.4. Note on Loss Metrics*

Although we specifically focused on accuracy assessment metrics and methods in this review, it is important to note that many of the considerations discussed above for accuracy assessment are also important for selecting an appropriate loss metric. The loss metric is the measure that is monitored during the training process. Over multiple learning epochs, or

iterations over the training data, the optimization algorithm updates weights to minimize or decrease the loss metric. Since loss should be reduced with improved performance, it is generally a measure of error as opposed to accuracy. Given that this metric serves to quantify error rates and guides the optimization algorithm, selecting an appropriate metric is very important [134].

For example, binary cross-entropy (BCE) loss, which is also referred to as log loss, often serves as the default loss metric for binary classification tasks [134–136]. However, it often performs poorly when data are imbalanced [134,137,138]. As a result, it may be more appropriate to apply class weighting, or weighted binary cross-entropy (WBCE), when classes are imbalanced [134,135]. Alternatively, 1—F1 Score or 1—Dice, generally referred to as Dice loss, is more robust to data imbalance than BCE [137–140]. It is even possible to combine multiple loss functions, such as BCE and Dice loss with equal or different weighting applied to each loss component [134]. Tversky loss augments the Dice loss to allow for different weighting to be applied to the FN and FP components, allowing the user to prioritize different types of error [134,141,142]. In the case of smooth Dice loss, class probabilities are used as opposed to the hard classification [134,143,144]. Focal loss, focal Dice loss, and focal Tversky loss augment BCE, Dice, and Tversky loss, respectively, to allow for increased focus on classifying difficult samples [139,142,145–147].

For multiclass classification, it is common to employ a multiclass version of cross-entropy (CE) loss. However, in cases of class imbalance, class weightings can be applied, a multiclass Dice loss can be used, or multiple losses can be combined, such as CE and multiclass Dice [134]. Loss metrics used for binary or multiclass classification are inappropriate for regression problems. In such cases, mean square error (MSE) or mean absolute error (MAE) are commonly used [148,149]. Models that have multiple outputs often use different loss metrics for each component, which can then be aggregated to a multi-task loss. For example, Mask R-CNN uses different loss metrics for classification, bounding box regression, and mask generation [97].

In summary, we encourage researchers to experiment with different loss functions and consider the impact of class imbalance and types of error. Just as it is important to select assessment metrics appropriate to the task being undertaken, choosing a loss function is also important, especially in the context of imbalanced class proportions. Ma et al. [134] provide a detailed discussion of loss functions and considerations for selecting loss functions in the context of medical image analysis, which is also relevant to RS studies.

## 5. Discussion and Recommendations

### 5.1. Comparison of DL and Traditional RS Approaches to Accuracy Assessment

Although the terminology used for accuracy assessment in RS DL papers is very different from that of traditional RS studies, the approaches have much in common, particularly in basing derived metrics on the confusion matrix. However, despite the importance of the confusion matrix, only 17% of the DL papers surveyed actually reported the entire confusion matrix. One reason for not presenting the complete confusion matrix is that in some studies, particularly scene classification, the number of categories is so large that confusion matrices simply take up too much space [35]. A complete confusion matrix is also normally not defined for object detection, because of a lack of an easily defined TN class. Nevertheless, it can be useful to provide a complete tabulation of the remaining three numbers of the object detection confusion matrix, TP, FP, and FN, to clarify the details of the accuracy evaluation.

The summary metric of OA is also common in both traditional RS and DL RS papers. Another similarity between traditional RS and DL RS accuracy is the common use of precision and recall, though traditional remote sensing names these metrics UA and PA respectively.

One of the major differences between traditional RS and DL RS is the use of F1 (which is a combination of precision and recall) and IoU, as well as mean F1 and mean IoU. Summary measures can be useful for ranking classifiers, but since multiple combinations of

precision and recall can produce the same F1 statistic or IoU, also reporting the underlying statistics, and, in particular, a population confusion matrix, can be helpful.

Several aspects of RS DL accuracy design stand out as being worthy of emulation by the broader RS community. Traditional RS studies could gain by using the P-R curve and the AP measure for classifications involving a threshold, for example, those using spectral ratios, such as the normalized difference snow index (NDSI) [150]. Moreover, the DL community in many cases designs the accuracy assessment to incorporate testing the algorithm in entirely new geographic regions, away from the data used in training, or images acquired on entirely different dates, or even with data acquired by different sensors. These testing designs give valuable insight regarding the generalization potential of algorithms and their likely performance in real-world monitoring applications, where it might be impractical to collect new training data to support every new data acquisition.

### 5.2. Clarity in Terminology

As discussed in Section 2.1, a primary purpose of accuracy assessment measures is to communicate the uncertainty associated with classification products. As the names of the RS DL accuracy measures used are mostly from the computer vision community, these terms may be a barrier to traditional remote sensing readers. For example, the traditional terms of UA and PA were reported in only 2% of the studies surveyed (Figure 3). Furthermore, the large number of alternate names used for each metric listed in Table 6 is a potential source of confusion to readers. When authors tabulate accuracy measures that though not identical are perfectly correlated (e.g., recall and false discovery rate [108] or F1 and IoU [151]), readers may not realize the information is redundant. Similarly, when studies refer to the same metric by different names in different parts of the paper (e.g., the text and tables [105]), or, in some cases, even in the same parts of the paper (e.g., within a single table [68]), communication may also be undermined. The problem is particularly acute when studies compare the ROC and P-R graphs, usually using true positive rate for the ROC, but recall for the same accuracy measure in the P-R graph [106,117,118,152]. Therefore, to the extent that it is possible, it would be preferable for studies to use the most common names in Table 6 (typically, the left column), rather than less common names. When multiple names are used for a single metric, highlighting the equivalency of the names would help readers.

Though it may seem a waste of space to define well-known metrics, such as most of those listed in Table 6, it is nevertheless useful when studies give the equations for the metrics used. Providing the equations will help RS readers less experienced in DL, as well as ensure there is no confusion as to exactly which measure is meant. This is particularly important for metrics that are given general names such as average accuracy or accuracy. Lack of consistency in the literature as to what is meant by some measures is not just limited to F1, AP, and mAP, but includes other terms, such as false alarm rate/probability (e.g., compare [100,103]), reinforcing the importance of this issue.

### 5.3. Class Prevalence and Imbalance

Some RS DL papers note concerns regarding class imbalance, and either implicitly or explicitly design the accuracy assessment to minimize the influence of class imbalance. Three main strategies are used. One approach is to use equalized stratified random sampling to generate a dataset with an equal number of samples in each class. This approach requires a subsequent analysis protocol to take into account that sampling does not reflect the prevalence of the class. Though many studies in the survey sampled each class equally, we identified no examples of studies that applied the necessary class prevalence correction. A second approach is to tabulate the values in the confusion matrix as percentages that sum to 100% in the reference categories, or occasionally, the predicted categories (Table 5). However, such a confusion matrix does not represent the population confusion matrix, but like the first approach, is instead, apparently, an attempt to represent a hypothetical situation in which each class has equal prevalence.

A third approach is to report only measures that are thought not sensitive to imbalance, such as recall or specificity (producer's accuracies in the terminology of traditional RS). Tharwat [98] provides an explanation of why these measures are thought not to be sensitive to imbalance. However, Foody has questioned the underlying assumption of this argument, stating, "... in common remote sensing applications the producer's accuracy may, however, be expected to be prevalent dependent" [12] p. 3). Perhaps more importantly, recall and precision are separate aspects of class accuracy, and it is entirely possible for one to be high, and the other low. Reporting only recall, and not precision, on the basis that the latter is sensitive to imbalance, will necessarily lead to only a partial understanding of classifier performance.

In summary, the three approaches to avoiding the issue of prevalence appear to be a denial that the prevalence of each class is an inherent feature of a classification, and that the class prevalence affects the accuracy of the classification outcome. Comparing classifier performance with classes of equal assumed prevalence means that the classifier is not tested with imbalanced classes, which is likely to be common in real remote sensing problems.

It could be argued that these concerns do not apply to scene classification applications, since they do not produce a map. However, as pointed out above, the proportion of the classes is a fundamental characteristic of all classifications, and thus is an important issue for all applications.

There are, however, a number of RS DL studies that, instead of trying to avoid imbalanced classes, incorporate prevalence as a factor in the accuracy assessment design. This trend is perhaps most significant in the development of benchmark datasets, because these datasets are often used in multiple studies. For example, Qian et al. [153] designed a change detection dataset of 3420 pairs of Google Earth images where the proportion of true change in the images varies from 0% to over 80%. Zhang et al. [68] designed their community synthetic aperture RaDAR (SAR) dataset for ship detection to have a large number of images where ships are rare, with a prevalence of just 0.0001%, unlike comparison datasets where pixels representing ships are between 2 and 4 orders of magnitude more prevalent. Another notable way the RS DL community has embraced the importance of considering prevalence is to move from the ROC to the P-R curve, the baseline of which provides information on the class prevalence.

## 6. Conclusions

We reviewed 100 randomly-selected papers focusing on DL classification that were published in eight major RS journals in 2020. As three of the papers each comprised two separate studies, this resulted in 103 studies in the survey. The review of these papers confirms that the RS DL community have largely abandoned traditional RS accuracy assessment terminology. The abandonment of traditional RS accuracy measures is most likely a result of the RS DL community drawing from the traditions of the computer science and AI communities. RS DL scientists use the metrics of those communities to facilitate communication of their work with the broader DL community, and a comparison of their work with other DL research outside that of RS applications. In addition, DL metrics such as F1 and IoU are seen as offering new ways of summarizing accuracy. The DL RS approach to accuracy assessment is grounded in the idea of the confusion matrix and measures that derive from it, but other than OA and Kappa, the names of the various measures tend to be different. The widely used DL terms of precision and recall are equivalent to UA and PA for the positive class in the case of a binary classification, or the class of interest for a multiclass classification.

There are several notable features of RS DL accuracy assessment compared to traditional RS studies. DL studies tend to, instead, average other statistics to produce a single class measure, most notably the F1 statistic, which is often described as the harmonic mean of precision and recall. It is, however, also conceptually similar to the IoU, though in the F1 measure, TP has twice the weighting it has in the IoU. In many cases, these individual class statistics are then averaged over all classes.

Another notable feature of RS DL studies is that, though the metrics used derive from the confusion matrix, they only rarely report a complete confusion matrix. For object detection, an important CNN application, the TN class is usually not defined, and so a complete confusion matrix is not possible. For scene classification, there may be so many classes that a complete confusion matrix may take up a lot of space. However, even for semantic segmentation, the complete confusion matrix is only rarely provided. Perhaps more significantly, we found only four studies where the confusion matrix was reported with numbers that were clearly defined as representing the proportion of that class in the landscape. Instead, when studies did report the confusion matrix, most present the entries as proportions, with either the reference data class or the predicted class values summing to 1.0. These types of individually class-normalized error matrices do not represent the population error matrix, and therefore, will not generate derived measures that represent unbiased estimates of the map accuracy. Instead, they appear to be an attempt to predict the accuracy of a hypothetical situation where the classes have equal prevalence. A key finding in our review of the RS DL literature is that the abandonment of traditional RS accuracy measures is not limited to a subset of RS journals, but is an almost universal change, and is consistent across all the journals surveyed. In addition, the issues that we highlight, including the use of different names for the same metric, the reporting of partially or completely redundant accuracy metrics, and the use of error matrices that do not necessarily represent an estimate of the population error matrix, also occur in all the journals surveyed.

Many RS DL studies are paying increased attention to the role of varying prevalence of classes in evaluating accuracy. Perhaps most importantly, this includes designing benchmark datasets that have class proportions that more closely reflect likely real-world scenes, or that provide scenes of varying and documented class proportions. Similarly, the dominant use of the P-R curve, along with the associated AP and/or mAP as an alternative to the traditional ROC and AUC, is also important in this regard. Although the P-R-curve, AP, and mAP are mainly used in object detection and semantic instance detection applications, these metrics are potentially useful for any classification involving a threshold.

Another important contribution of RS DL studies is to emphasize the testing of trained models using entirely new datasets, for example, data of a different region, a different time, or acquired from a different sensor. This type of experimental accuracy assessment design points to the potential of RS for monitoring applications, and could well be followed by RS studies investigating traditional classification methods. In the next paper in this series (Maxwell et al., in review), we provide recommendations for best practices for assessing CNN-based products based on the findings from this literature review.

**Author Contributions:** Conceptualization, A.E.M. and T.A.W.; formal analysis, T.A.W. and L.A.G.; investigation, A.E.M., T.A.W., and L.A.G.; data curation, A.E.M.; writing—original draft preparation, A.E.M., T.A.W., and L.A.G.; writing—review and editing, A.E.M., T.A.W., and L.A.G. All authors have read and agreed to the published version of the manuscript.

**Funding:** This work was funded by the National Science Foundation (NSF) (Federal Award ID Number 2046059: "CAREER: Mapping Anthropocene Geomorphology with Deep Learning, Big Data Spatial Analytics, and LiDAR").

**Institutional Review Board Statement:** Not applicable.

**Informed Consent Statement:** Not applicable.

**Acknowledgments:** We would also like to thank three anonymous reviewers whose comments strengthened the work.

**Conflicts of Interest:** The authors declare no conflict of interest.

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
