# Peer review of "Accuracy Assessment in Convolutional Neural Network-Based Deep Learning Remote Sensing Studies—Part 1: Literature Review"

_remotesensing, doi:10.3390/rs13132450_

Round 1

Reviewer 1 Report

This review manuscript is well-written and is recommended for acceptance. The manuscript discusses the trend of traditional Accuracy Assessment methods and Accuracy Assessment methods for deep learning algorithm when deep learning technology is gradually applied to remote sensing. But some comments should also be proposed.

Comment1: The quantity of literatures selected by the authors is enough, however, the quality of the paper is not mentioned. Judging from the influence of the journal in which the paper is accepted, and the number of paper’s citations, Does the accuracy assessment method used in this paper have a certain trend?This may need to be mentioned.

Comment2: line 129-line 133, There should be references to support this.

Comment3: line 133-line 136, There should be references to support this.

Comment4: In the manuscript, the author discusses the application of DL in RS, and uses many DL's accuracies, but many of them are bound to the technology adopted (such as semantic segmentation and mean intersection over union, paired images and PSNR and SSIM), which should be further explained

Comment5: line 405 (Figure 3. Graph of F1 vs. IoU for 100 randomly generated points, illustrating the correlated nature 406), the parameters represented by the points, lines and circles in the graph should be explained in the graph

Comment6: for Figure 4, the text used for interpretation is too large, for Figure 5, the text used for interpretation is too large. For Figure 6, the text used for interpretation is too large, the font of image number is obviously inconsistent with the main body, for Figure 7, the font of image number is obviously inconsistent with the main body, in the Figure (d), the line is too thick.

Comment7: The author should explain the difference between traditional RS and DL-based methods to better explain the reasons for the changes in the Accuracy Assessment method.

Comment8: As a review type manuscript, the author should discuss the reasons and the tendency for the changes or in the Accuracy Assessment method

Comment9: As a review paper, after the summary, there should better be a separate chapter for discussion

Comment10: I suggest that the author supplement the prospect of remote sensing technology based on deep learning, and combine the development trend of Accuracy Assessment method

Comment11: In Figure 2, some Accuracy Assessment methods are obviously not applicable to the corresponding deep learning methods. Should we consider the authenticity of these articles? I suggest that this table should be embodied in the form of digitized percentages, and the methods that are not applicable should be marked

Reviewer 2 Report

In general, I think that this study presents a useful step forward for the community and that it builds onto the present state of knowledge. I have a couple of concerns with the presented introduction and methodology that should be addressed. Also, you need to provide more CNN-related literature reviews in the introduction section associated with the research gap/limitation. I am providing a few, but you will get more literature related to CNN evaluation and error correction.  Otherwise, the readers cannot see the importance of your study.

 My major comments and questions are as follows:

  • A recent study highlighted the importance of the band combinations in the use of multispectral datasets on deep learning convolutional neural net (DLCNN) model prediction accuracy for remote sensing application. The authors should explain this aspect in the introduction section.

Abdalla, Alwaseela, et al. "Color Calibration of Proximal Sensing RGB Images of Oilseed Rape Canopy via Deep Learning Combined with K-Means Algorithm." Remote Sensing 11.24 (2019): 3001.

Park, Ji Hyun, et al. "RGB Image Prioritization Using Convolutional Neural Network on a Microprocessor for Nanosatellites." Remote Sensing 12.23 (2020): 3941.

Yang, Ronglu, et al. "Representative band selection for hyperspectral image classification." Journal of Visual Communication and Image Representation 48 (2017): 396-403.

  • You can add a couple of paragraphs explaining high-resolution remote sensing applications as well as the transferability of the Neural network structure for remote sensing applications? How did recent study explore the geographic transferability based on deep learning convolutional neural net?

Yan, D.; Li, G.; Li, X.; Zhang, H.; Lei, H.; Lu, K.; Cheng, M.; Zhu, F. An Improved Faster R-CNN Method to Detect Tailings Ponds from High-Resolution Remote Sensing Images. Remote Sens. 2021, 13, 2052. https://doi.org/10.3390/rs13112052

  • Circle-U-Net-based Efficient deep learning convolutional neural net architecture for Semantic Segmentation is one of the popular new approaches in remote sensing approach. You need to introduce this aspect in this review works.

Sun, F.; V, A.K.; Yang, G.; Zhang, A.; Zhang, Y. Circle-U-Net: An Efficient Architecture for Semantic Segmentation. Algorithms 2021, 14, 159. https://doi.org/10.3390/a14060159

Ronneberger, O.; Fischer, P.; Brox, T. U-Net: Convolutional Networks for Biomedical Image Segmentation. arXiv 2015, arXiv:1505.04597.

  • Data fusion, the process of combining multispectral (MS) and high-resolution panchromatic (PAN) images with complementary characteristics often serve as an integral component of remote sensing mapping workflows. The fusion process generates spectral and spatial artifacts that affect the classification accuracies of subsequent automated image analysis algorithms, such as deep learning (DL) convolutional neural nets (CNN). A recent study suggested that the DL-based fusion algorithms that preserve the spatial character of original PAN imagery favor the DLCNN model performances in order to enable an accurate mapping effort in remote sensing applications. You did not discuss anything in detail about the fusion algorithm. You should explain these issues? (See: Witharana et al. 2020; Shahdoosti, 2017 You will get some insight and you can discuss it)?

Witharana, C.; Bhuiyan, M.A.E.; Liljedahl, A.K.; Kanevskiy, M.; Epstein, H.E.; Jones, B.M.; Daanen, R.; Griffin, C.G.; Kent, K.; Jones, M.K.W. Understanding the synergies of deep learning and data fusion of multispectral and panchromatic high resolution commercial satellite imagery for automated ice-wedge polygon detection. ISPRS J. Photogramm. Remote Sens. 2020, 170, 174–191.

  1. Shao and J. Cai, "Remote Sensing Image Fusion With Deep Convolutional Neural Network," in IEEE Journal of Selected Topics in Applied Earth Observations and Remote Sensing, vol. 11, no. 5, pp. 1656-1669, May 2018, doi: 10.1109/JSTARS.2018.2805923.

  • Can you explain the high-resolution features and high-level semantic information with examples?
  • How about loss function in DL-based remote seeing application? Authors should provide a detailed discussion about loss function along with few feature maps which will provide a clear idea about the DL structure.
  • Also, You should provide a discussion about the impact of hyperparameter settings for different remote sensing applications.
  • Can you explain overfitting issues that are completely missing?

Reviewer 3 Report

The work reviews metrics for accuracy assessment, including analysis of a selection of 100 papers. Increasing consistency in the literature regarding use of metrics, which will make papers more intercomparable, increase reading understanding, etc. I think the work is interesting and very well written. A few trivial comments:

Related to terminology, Type I and II errors are also commonly referred across fields. I suggestion adding mention of them

L515-516 The questioning of kappa has been mentioned many times at this point in the manuscript. This line feels repetitive and could be deleted

L660 ‘think’ => ‘thank’

Ref 13 first author’s name (Pontius) is cut off

Ref 30 is published - https://ieeexplore.ieee.org/document/9127795

Ref 67 is published - https://ieeexplore.ieee.org/document/7803544

Ref 74 – citation seems to be incomplete - https://ieeexplore.ieee.org/document/8372616

Ref 75 is no longer ahead-of-print; also should add [75] when mentioned on L335

Ref 93 is published - https://ieeexplore.ieee.org/abstract/document/9079478

Ref 101 – citation is incomplete

Round 2

Reviewer 1 Report

Thank you for adopting my revised opinion

Reviewer 2 Report

The authors significantly improved the quality of the paper by addressing most of the comments.

In the introduction, the section can you add one more paragraph for the High-Resolution Remote Sensing Images application using the recently established state of the arts deep learning technique where they utilized optimal band combinations (e.g. three-band combinations) and exhibited significant accuracy in remote sensing applications. They successfully developed automatic extraction framework for remote sensing applications from high spatial resolution optical images using CNN architecture in a large-scale application based on multispectral band combinations. Please introduce these latest advanced research works, and their potential impact, and their limitation in terms of algorithms. It would help the manuscript tremendously if the state-of-the-art was more streamlined in the introduction section. Recent study results highlighted the importance of the band combinations in the use of multispectral datasets on model prediction accuracy for remote sensing application (Cai, et al.2021; Park et al.2020; Ehsan, et al.2020; Li, et al. 2018; Abdalla, Alwaseela, et al.2019; Yanget al 2019). The authors should explain this aspect in the introduction section.

I am providing below studies which will be perfectly fit in your review paper:

Li, Yuqi, et al. "Optimized multi-spectral filter array-based imaging of natural scenes." Sensors 18.4 (2018): 1172.

Ehsan et al. 2020 “Understanding the Effects of Optimal Combination of Spectral Bands on Deep Learning Model Predictions: A Case Study Based on Permafrost Tundra Landform Mapping Using High-Resolution Multispectral Satellite Imagery. J. Imaging 2020, 6, 97.”

Abdalla, Alwaseela, et al. "Color Calibration of Proximal Sensing RGB Images of Oilseed Rape Canopy via Deep Learning Combined with K-Means Algorithm." Remote Sensing 11.24 (2019): 3001.

Park, Ji Hyun, et al. "RGB Image Prioritization Using Convolutional Neural Network on a Microprocessor for Nanosatellites." Remote Sensing 12.23 (2020): 3941.

Yang, Ronglu, et al. "Representative band selection for hyperspectral image classification." Journal of Visual Communication and Image Representation 48 (2017): 396-403.

Cai, Y.; Huang, H.; Wang, K.; Zhang, C.; Fan, L.; Guo, F. Selecting Optimal Combination of Data Channels for Semantic Segmentation in City Information Modelling (CIM). Remote Sens. 2021, 13, 1367. https://doi.org/10.3390/rs13071367

If you find more related literature please add to your revised version of the manuscript.
